# Qualitative Evaluation of Intracranial Pressure Slopes in Patients Undergoing Brain Death Protocol

**DOI:** 10.3390/brainsci13030401

**Published:** 2023-02-25

**Authors:** Mylena Miki Lopes Ideta, Louise Makarem Oliveira, Daniel Buzaglo Gonçalves, Mylla Christie de Oliveira Paschoalino, Nise Alessandra de Carvalho Sousa, Marcus Vinicius Della Coletta, Wellingson Paiva, Sérgio Brasil, Robson Luís Oliveira de Amorim

**Affiliations:** 1Department of Internal Medicine, Federal University of Amazonas, Manaus 69077, ZIP, Brazil; 2Department of Neurology, University of São Paulo, Ribeirão Preto 14049, ZIP, Brazil; 3Department of Neurology, Amazonas State University, Manaus 69005, ZIP, Brazil; 4Division of Neurosurgery, Department of Neurology, University of São Paulo, São Paulo 01246, ZIP, Brazil

**Keywords:** brain death diagnosis, intracranial pressure, intracranial compliance

## Abstract

Background: Due to the importance of not mistaking when determining the brain death (BD) diagnostic, reliable confirmatory exams should be performed to enhance its security. This study aims to evaluate the intracranial pressure (ICP) pulse morphology behavior in brain-dead patients through a noninvasive monitoring system. Methods: A pilot case-control study was conducted in adults that met the BD national protocol criteria. Quantitative parameters from the ICP waveforms, such as the P2/P1 ratio, time-to-peak (TTP) and pulse amplitude (AMP) were extracted and analyzed comparing BD patients and health subjects. Results: Fifteen patients were included, and 6172 waveforms were analyzed. ICP waveforms presented substantial differences amidst BD patients when compared to the control group, especially AMP, which had lower values in patients diagnosed with BD (*p* < 0.0001) and the TTP median (*p* < 0.00001), but no significance was found for the P2/P1 ratio (*p* = 0.8). The area under curve for combination of parameters on the BD prediction was 0.77. Conclusions: In this exploratory study, noninvasive ICP waveforms have shown potential as a screening method in patients with suspected brain death. Future studies should be carried out in a larger population.

## 1. Introduction

The neurological determination of death, or brain death (BD), must follow rigorous medical standards in order to not allow mistakes [1,2,3]. Although in some countries the diagnosis of BD is given solely by clinical examination, in others, such as in Brazil, ancillary exams are necessary to complement the neurological assessment [4,5]. Therefore, examination methods which can aid on the diagnostic process have been widely studied, being the most used techniques the transcranial Doppler (TCD), computed tomography angiography and electroencephalogram [6,7,8].

ICP monitoring is also considered a complementary examination in BD diagnosis, since if ICP overreaches mean arterial pressure, cerebral perfusion pressure (CPP) is null [6]. However, this technique is an invasive procedure, is costly and is restricted to some cases of acute brain injuries. Moreover, it is associated with complications and risks, mainly infection, hemorrhage and obstruction [9,10,11]. As BD is a condition of perfusion collapse because of extremely high ICP, one rationale on its monitoring could be the ICP waveforms (ICPW), considering that ICPW change according to reduction in intracranial compliance [12].

Recently, the development of a non-invasive system to evaluate the ICPW (nICPW) allowed assessing the behavior of ICPW beyond the neurointensive care environment. This method has clinically [9,13,14] and experimentally [15,16] demonstrated a high correlation with its invasive predicate, reproducing the same pulse shape profile and extracting numerical parameters from its different peaks amplitudes. Therefore, this pilot study aims to evaluate the behavior of nICPW in patients with a confirmed diagnosis of BD. Our specific goal was to analyze the quantitative differences between ICPW features comparing health individuals and patients undergoing brain death protocol.

## 2. Material and Methods

### 2.1. Study Design

The study is an exploratory and analytical pilot case-control study. It was carried out in Getulio Vargas University Hospital and João Lucio Hospital, in Manaus, Brazil. In compliance with the ethical aspects and the requirements, this study was approved by the Amazonas Federal University Research Ethics Committee, under the registration number: 82714517.2.0000.5020.

### 2.2. Neuromonitoring

nICPW were assessed using the B4C (B4C; Brain4care Corp., São Carlos, Brazil) sensor, which consisted into a monitor that quantifies local cranial bone deformations using specific sensors [17]. In summary, the system is able to capture micrometric skull deformations according to ICP variation, and the influence of the cardiac cycle represents the pulse slopes of the ICP waveform. The system was positioned in the frontotemporal region, approximately 3 cm over the first third of the orbitomeatal line, at the same side of ICP catheter implantation. Consequently, the main branches of the temporal superficial artery and the temporal muscle were avoided, and sensor contact was provided through an area of thin skin and skull bone, whereas slight pressure was applied to the adjustable band until an optimal signal was detected. Through an artificial intelligence processing, numerical parameters are derived from the waveforms, as pulse amplitude (AMP), P2/P1 ratio and time-to-peak (TTP), which change in accordance with ICC exhaustion [12]. The sampling frequency is 200 Hz; the cloud algorithm processes multiple heartbeats and generates a minute-by-minute report with the minute average of P2/P1 ratio and TTP.

### 2.3. Inclusion Criteria

Adults (18 years old and older) of any gender and admission diagnostic, with a positive BD examination following the Brazilian guidelines for BD determination [18] (a clear reason for coma documented, absence of sedation, absence of brain stem reflexes and apnea test) were included. TCD was used as the ancillary technique to indicate the intracranial blood circulatory arrest. TCD patterns admitted as indicators of BD were systolic spikes up to 50 cm/s or the reverberating oscillatory flow [19]. In the control group, healthy volunteers not submitted to sedation, without signs or symptoms of intracranial hypertension or diagnosis of previous neurological disease that could be responsible for altered ICP were included. These individuals were matched for gender and age with the participants in the case group.

### 2.4. Exclusion Criteria

Patients who underwent craniectomy and patients suffering from perforating or penetrating skull injuries were excluded.

### 2.5. Data Collection

Data collection was performed for 5 min with the individual in a supine position with the head elevated to 30 degrees. TCD examinations were performed following B4C assessments. Physiological and epidemiological data were collected. Parameters derived from nICPW were collected and systematically analyzed offline.

### 2.6. Statistical Analysis

Continuous variables were presented as mean and standard deviation using a 95% confidence interval. The Wilcoxon test was used for comparisons between continuous variables in non-normal distribution. The waveforms were sorted in case (9 patients) and control (6 healthy subjects) groups. Seven models using different combinations of the parameters derived from nICPW (Table 1) were adjusted by logistic regression, establishing the area under the curve (AUC). All statistics were performed based on the number of pulses assessed for each group by an independent statistician. The data separation and adjustment were performed 500 times. *p* < 0.05 was considered statistically significant, and all tests were two-tailed. Data were analyzed using the R software (version 4.2.2, available at https://www.r-project.org/ (accessed on 1 December 2022)).

## 3. Results

Among 15 subjects included, the average age for controls (50% female) was 44.83 years ranging from 21 to 53, whereas for the case group (44% female); the average age was 48.5 years, ranging from 21 to 61. The underlying diseases of these patients were: hemorrhagic stroke (4), brain neoplasms (1) and closed head injury (1). The complimentary exam performed for brain death diagnosis was the TCD in all brain-dead patients concomitantly with the nICPW system.

Regarding the nICPW analyses, we achieved 6172 observations obtained pulse to pulse, including both control (2297 waveforms) and case (3875 waveforms) groups. Pulse amplitude had lower values in patients diagnosed with BD (*p* < 0.0001) (Figure 1a). In contrast, the P2/P1 ratio median was not significant (*p* = 0.80). TTP medians’ comparison revealed significant differences amidst case and control groups (*p* < 0.00001) (Figure 1c).

Joint correspondence analysis of all the observations obtained from the data collection, case group and control group is discriminated in Figure 2. Regarding the models tested for validation, model 3 (pulse amplitude) obtained the best performance since it presented the greater average AUC, which value corresponded to 0.77 (Confidence interval [CI] = 2.076; 2.439) (Figure 3). For this model, defining the confusion matrix to obtain statistical significance depending on the chosen parameter (AMP) rather than the sample size, the cut-off was predefined in 0.001, and then the results were distributed between cases and controls, with 58% sensitivity, 66% specificity and 60% accuracy.

## 4. Discussion

This exploratory study has shown the possibility of the nICPW to be statistically different among healthy and brain-dead subjects. However, our preliminary analysis led to a sole moderate accuracy, with consequences that may not be extrapolated to the clinical practice yet, as a complimentary exam to testify BD. Three parameters derived from nICPW were assessed, the pulse amplitude, P2/P1 ratio and the time to peak. These parameters have scientific bases previously reported [13,20,21], being the literature the premise to perform the present analyses. Our modest results may be explained because in BD, despite an extreme level of intracranial hypertension having been reached [6], the absence of beat-by-beat variation on intracranial volume impacts the ICP waveform. Moreover, this can explain the particular impact we observed in our study over the P2/P1 ratio, since the P2 is translated as the tidal wave, that is, the spread of blood thru the brain after maximum arterial ejection (ending of systolic phase) [22].

Scherzer et al. [23] refer to ICP monitoring as an investigation method useful in the early timing, in order not to delay the diagnosis of BD and its medical consequences. Invasive ICP evaluation in BD was described as reaching a peak maximum in 5 to 12 h, subsequently initiating a wave amplitude decrease [24]. All of the 31 BD diagnosed patients who had ICP levels analyzed by Salih et al. [25] presented severely elevated ICP at the moment of diagnosis (95.5 mmHg ± 9.8 mmHg), as well as reduced CPP. Roth et al. [26] performed retrospective data of 18 patients with ICP monitoring during the development of BD due to primary brain lesions. All patients in this study experienced ICP values > 95 mmHg and CPP < 10 mmHg. Interestingly, the ICPW obtained from invasive ICP analysis has been already described in BD. Domínguez et al. [27] identified a P2-predominant pattern, although not specific to BD, once observed in both BD and survivors groups. The disappearance of P3 in most BD diagnosed patients was associated with the compromise of cerebral venous outflow, suggesting the presence of severe disturbance of blood flow in this pathology. There is agreement between these observations with the classic study from Nucci et al. that observed that P3 would become of higher amplitude than p1 in severe states of intracranial hypertension, and the consequence is a pyramidal pulse shape [28].

However, to our knowledge, no studies performed with invasive ICP monitoring focused on the ICPW quantitative data, mainly because of the difficulties in extracting these parameters quantitatively. Rozsa et al. [29] compared ultrasonic pulse wave assessment in eight BD patients and 34 neurologically healthy volunteers. Although not specific enough to use it as a criterion for BD, the patient group waveform was typical. Sub-wave values for P1, P2 and P3 were obtained, and intracranial pulses amplitude were significantly lower than controls, similar to what we have found in the present study. In BD, the P1 peak is low because there is no rapid increase in cerebral blood volume or consequent rapid increase in cerebrospinal fluid volume. In contrast, P2 and P3 peaks may stay high since there is no rapid venous outflow [30]. However, in our data, we could not see an increase in the P2/P1 ratio, probably because ICP has reached an extremely high value and no more intracranial volume changes are seen during the cardiac cycle.

This study is the first to evaluate nICPW in BD patients. It is considered a new bedside strategy that provides real-time monitoring. ICP pulse morphology presented substantial differences amid BD patients when compared to the control group. The pulse amplitude and TTP variables values in the case group were statistically significant. Hence, this new nICPW monitoring device may be a useful screening tool to identify possible brain-dead patients. Future studies may search for more accurate additional parameters to extract from nICPW, assessing also its behavior according to arterial pressure variations.

## 5. Limitations

This study has several limitations. Firstly, the reduced sample size may underpower this study. Then, a better, accurate model was prevented. However, as this is an exploratory study, it gave some insights to be used in a larger study. Our second main limitation is the absence of patients with a different spectrum of consciousness disorders or sedated patients. No specific limitations regarding data collection and analysis are reported, since the preclusion to handle the B4C system and acquire its waveforms is agitation, which was not an issue for the sample of the present study.

## 6. Conclusions

In this exploratory study, parameters extracted from noninvasive ICP waveform as pulse amplitude and the time interval for highest peak amplitude were significantly different between brain-dead patients and healthy subjects. Further studies with larger samples may determine the role of this technique on the brain death assessment complementation.

## Figures and Tables

**Figure 1 brainsci-13-00401-f001:**
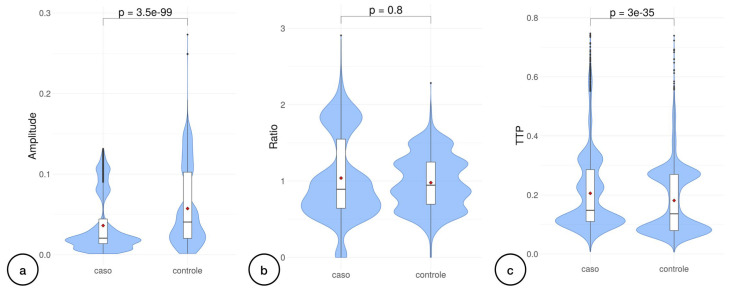
Statistical analysis by Wilcoxon–Mann–Whitney of the variables: pulse amplitude (**a**); P2/P1 ratio (**b**); TTP (**c**).

**Figure 2 brainsci-13-00401-f002:**
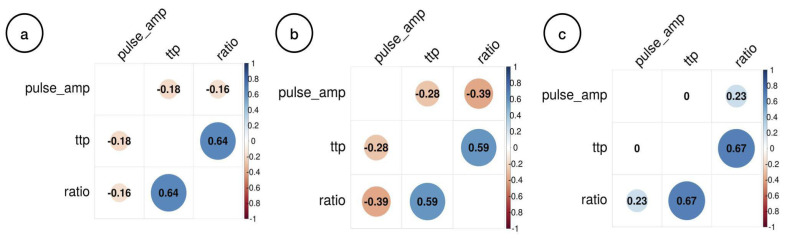
Joint Correspondence analysis of all the observations obtained from the data collection (**a**), case group (**b**), and control group (**c**). pulse_amp = Pulse amplitude; ttp = Time to peak; ratio = P2/P1 ratio.

**Figure 3 brainsci-13-00401-f003:**
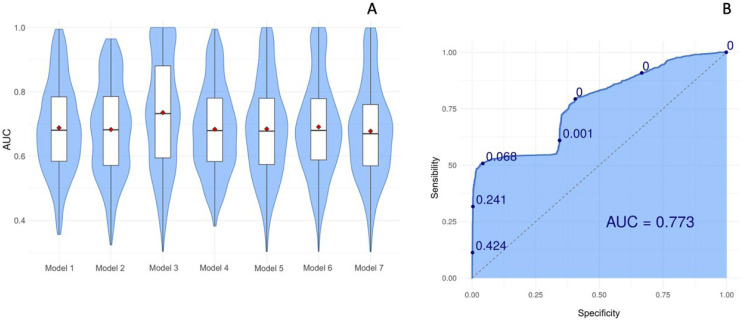
AUC calculated according to the model adjusted by logistic regression. (**A**) Model 3 presents a superior AUC, therefore obtained the best performance in this comparison. (**B**) The graph represents AUC calculated for Model 3, based on sensitivity and specificity values.

**Table 1 brainsci-13-00401-t001:** Analytic model descriptions used to calculate the best area under curve for nICPW parameter(s). TTP: time to peak, pulse_amp: pulse amplitude.

Model 1	Group~P2/P1 ratio
Model 2	group~ttp
Model 3	group~pulse_amp
Model 4	group~P2/P1 ratio + ttp
Model 5	group~ttp + pulse_amp
Model 6	group~P2/P1 ratio + pulse_amp
Model 7	Group~P2/P1 ratio + ttp + pulse_amp

## Data Availability

The datasets generated during and/or analyzed during the current study are available from the corresponding author on reasonable request.

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
