# Peer review of "Qualitative Evaluation of Intracranial Pressure Slopes in Patients Undergoing Brain Death Protocol"

_brainsci, 2023, doi:10.3390/brainsci13030401_

Round 1

Reviewer 1 Report

Well structured , well written presentation of the study conducted by the authors. Though some details of the system used would be usefull to be repeated in the present paper; I don' t think it change the overall quality of the article.

Author Response

Reviewer 1

Well structured, well written presentation of the study conducted by the authors. Though some details of the system used would be usefull to be repeated in the present paper; I don' t think it change the overall quality of the article.

Thank you for the positive comments, we added a more comprehensive description of the system, please check it.

Reviewer 2

  1. The article title should mention the manuscript type.
  2. Abstract
  3. Provide more percentages and statistical significance in the description of the results. It is advised to include more quantitative data.

Thank you for this comment, it was added, please check.

  1. Inclusion criteria
  2. Provide a reference for the brain death protocol

Added at page 4

  1. Statistics
  2. How was calculated the power of the study?
  3. Describe the distribution of the variables
  4. How were confounding variables assessed?
  5. Include the place where it was developed STATA.

The statistical analysis section was rewritten, please check it.

  1. Limitations
  2. A description of specific limitations to data collection and analysis should be done.

Thank you, it was made clearer in the corresponding session, please check it.

The Reviewer would like to ask, “how did the authors exclude confounding variables specifically related to the waveform of TCD?” It is well-known that the wave could variate between different subjects. How did the authors select specific sonology characteristics and evaluate them?

Thank you for the comment. The TCD patterns have been better described on page 4. Systolic spikes and the oscillatory flow are accepted worldwide as the TCD spectra for BD.

Reviewer 3

The paper describes the assessment of intracranial pressure slopes in patients undergoing the brain death protocol. The paper is interesting, but in my opinion, some concerns need to be addressed before publication:

1)      The Material and Methods section states that the instrumentation's specifics are disclosed elsewhere, but in my opinion, details like the sampling frequency and signal preprocessing should be included in the manuscript.

Thank you, the material and methods section has been improved, please check it.

2)       In the results section, it is stated that 6,172 observations, pulse to pulse, were obtained (line 96). Please provide some more details regarding the distribution of these samples (how many controls and how many brain-death patients). Why was the Wilcoxon test employed rather than the t-test? Was the distribution not normal? Please provide some details.

The distribution was not normal, 3875 waveforms were analyzed from the 9 BD patients and 2297 waveforms were analyzed from the controls.

3)      Please provide some information regarding the models employed, explaining the differences between models from 1 to 7. 

Table 1 has been provided to explain better the models applied.

4)      Although it is stated that "the waveforms were sorted in training (10 patients) and validation (5 patients)" (line 83, page 2), the sample size for each class is not provided. In fact, it is assumed that the models were fed using the characteristics derived on the 6,172 participant observations. Please share additional details regarding the development of the models.

Indeed all stats were made according to the number of waveforms instead of patients, but they always separated in case and controls. Table 1 was provided to clarify this point and the statistical analysis was rewritten, please check it.

5)      In my opinion, given the limited number of participants, leave-one-subject-out cross-validation could be advantageous since, with this approach, the training set is the largest possible. Moreover, no information regarding the balance of the two classes is reported. This aspect should be specified in the manuscript since employing classes that are not balanced could introduce a possible overfitting effect. In my opinion, in order to reduce the risk of overfitting effects, an iterative procedure that investigates all the possible compositions of the training and test sets should be performed in order to limit the dependence of the results on the study sample. However, it should be noted that in Figure 3, the mean AUC and the standard deviation seem to be reported; hence, maybe an approach similar to the one that I proposed has been applied. Importantly, Figure 3 shows a great variability of the results, highlighting a dependence on the study sample. Please provide some detail about the classification task and discuss this aspect in the Discussion section.

Thank you for the comment. We realize that the parameters used in this assessment are limited to discriminate BD. The multiple models created revealed only modest results, thus, next explorations of ICPW must focus on new and additional parameters, and this will be the aim of future works. We added this in the discussion. Please Check it.

Reviewer 2 Report

1.     The article title should mention the manuscript type.

2.     Abstract

a.     Provide more percentages and statistical significance in the description of the results. It is advised to include more quantitative data.

3.     Inclusion criteria

a.     Provide a reference for the brain death protocol

4.     Statistics

a.     How was calculated the power of the study?

b.     Describe the distribution of the variables

c.     How were confounding variables assessed?

d.     Include the place where it was developed STATA.

5.     Limitations

a.     A description of specific limitations to data collection and analysis should be done.

The Reviewer would like to ask, “how did the authors exclude confounding variables specifically related to the waveform of TCD?” It is well-known that the wave could variate between different subjects. How did the authors select specific sonology characteristics and evaluate them?

There are some grammatical errors throughout the manuscript that should be addressed. E.g.,

-       L12 abatract – abstract

-       Revise the address of every author.

Author Response

(The authors gave the same response as above.)

Reviewer 3 Report

The paper describes the assessment of intracranial pressure slopes in patients undergoing the brain death protocol. The paper is interesting, but in my opinion, some concerns need to be addressed before publication:

1)      I recommend transferring the study sample description (line 89 page 3) from the results section to the Material and Methods section.

The Material and Methods section states that the instrumentation's specifics are disclosed elsewhere, but in my opinion, details like the sampling frequency and signal preprocessing should be included in the manuscript.

2)       In the results section, it is stated that 6,172 observations, pulse to pulse, were obtained (line 96). Please provide some more details regarding the distribution of these samples (how many controls and how many brain-death patients). Why was the Wilcoxon test employed rather than the t-test? Was the distribution not normal? Please provide some details.

3)      Please provide some information regarding the models employed, explaining the differences between models from 1 to 7.

4)      Although it is stated that "the waveforms were sorted in training (10 patients) and validation (5 patients)" (line 83, page 2), the sample size for each class is not provided. In fact, it is assumed that the models were fed using the characteristics derived on the 6,172 participant observations. Please share additional details regarding the development of the models.

5)      In my opinion, given the limited number of participants, leave-one-subject-out cross-validation could be advantageous since, with this approach, the training set is the largest possible. Moreover, no information regarding the balance of the two classes is reported. This aspect should be specified in the manuscript since employing classes that are not balanced could introduce a possible overfitting effect. In my opinion, in order to reduce the risk of overfitting effects, an iterative procedure that investigates all the possible compositions of the training and test sets should be performed in order to limit the dependence of the results on the study sample. However, it should be noted that in Figure 3, the mean AUC and the standard deviation seem to be reported; hence, maybe an approach similar to the one that I proposed has been applied. Importantly, Figure 3 shows a great variability of the results, highlighting a dependence on the study sample. Please provide some detail about the classification task and discuss this aspect in the Discussion section.

Author Response

(The authors gave the same response as above.)

Round 2

Reviewer 3 Report

I thank the Authors for addressing my concerns. I have only one last consideration. Given the unbalanced classes (3875 waveforms from patients and 2297 from controls), the accuracy could depend on the capability of the model to classify the most numerous class; hence, please report the confusion matrix associated with the model with the best performance in order to make explicit the independence of the performance from the study sample.

Author Response

Thank you very much for this important observation.

All stats were done by an independent statistician, "blinded" to the situation of the subjects included. After assessing and distributing the results according to predefined cut-offs, the observations were assigned to the right group they belonged to.

The paragraph added is as follows:

"For this model, defining the confusion matrix to obtain statistical significance depending on the chosen parameter (AMP) rather than the sample size, the cut-off was predefined in 0.001 and then the results were distributed between cases and controls, with 58% sensitivity, 66% specificity and 60% accuracy."   I hope this can satisfy your demand.   With kind regards,